# The Importance of Natural and Acquired Immunity to SARS-CoV-2 Infection in Patients on Peritoneal Dialysis

**DOI:** 10.3390/vaccines12020135

**Published:** 2024-01-29

**Authors:** Marko Baralić, Mirjana Laušević, Danica Ćujić, Ana Bontić, Jelena Pavlović, Voin Brković, Aleksandra Kezić, Kristina Mihajlovski, Lara Hadži Tanović, Iman Assi Milošević, Jovana Lukić, Marija Gnjatović, Aleksandra Todorović, Nikola M. Stojanović, Dijana Jovanović, Milan Radović

**Affiliations:** 1Faculty of Medicine, University of Belgrade, Doktora Subotića Starijeg 8, 11000 Belgrade, Serbia; mirjana.lausevic@med.bg.ac.rs (M.L.); jelena@pavlovic.rs (J.P.); voin.brkovic@med.bg.ac.rs (V.B.); lukic_jovana@yahoo.com (J.L.); dijanaj@eunet.rs (D.J.); milan.radovic@med.bg.ac.rs (M.R.); 2Clinic of Nephrology, University Clinical Centre of Serbia (UCCS), Pasterova 2, 11000 Belgrade, Serbia; larahadzitanovic@gmail.com (L.H.T.); iman.assi@yahoo.com (I.A.M.); 3Institute for the Application of Nuclear Energy (INEP), University of Belgrade, Banatska 31b, 11080 Belgrade, Serbia; danicac@inep.co.rs (D.Ć.); marijad@inep.co.rs (M.G.), aleksandra.todorovic@inep.co.rs (A.T.); 4Department of Environmental and Occupational Health, University of Nevada, Las Vegas, NV 89154, USA; mihajk1@unlv.nevada.edu; 5Department of Physiology, Faculty of Medicine, University of Niš, Bulevar Zorana Đinđića 81, 18000 Niš, Serbia; nikola.stojanovic@medfak.ni.ac.rs

**Keywords:** peritoneal dialysis, SARS-CoV-2, vaccines, antibodies, effluent

## Abstract

The pandemic caused by the SARS-CoV-2 virus had a great impact on the population of patients treated with peritoneal dialysis (PD). This study demonstrates the impact of infection and vaccination in 66 patients treated with PD and their outcomes during a 6-month follow-up. This is the first research that has studied the dynamics of anti-SARS-CoV-2 IgG in serum and effluent. In our research, 57.6% of PD patients were vaccinated, predominantly with Sinopharm (81.6%), which was also the most frequently administered vaccine in the Republic of Serbia at the beginning of immunization. During the monitoring period, the level of anti-SARS-CoV-2 IgG antibodies in the PD patients had an increasing trend in serum. In the group of vaccinated patients with PD, anti-SARS-CoV-2 IgG antibodies had an increasing trend in both serum and effluent, in contrast to non-vaccinated patients, where they decreased in effluent regardless of the trend of increase in serum, but statistical significance was not reached. In contrast to vaccinated (immunized) patients who did not acquire infection, the patients who only underwent the COVID-19 infection, but were not immunized, were more prone to reinfection upon the outbreak of a new viral strain, yet without severe clinical presentation and with no need for hospital treatment.

## 1. Introduction

Peritoneal dialysis (PD) is one of the three complementary methods of kidney replacement therapy (KRT) which treats almost 300,000 people with end-stage kidney disease (ESKD) worldwide, representing 11% of the total dialysis population [1]. It is significantly less prevalent than the widespread hemodialysis (HD), both in the world and in the Republic of Serbia [2,3]. The peritoneal membrane has various pores, and large pores with a radius of 250 Å, which play a role in the transcapillary transport of macromolecules, such as proteins and immunoglobulins (Ig), by the process of active transport, and represent only 0.01% of the total [4]. During the period of the COVID-19 pandemic, a small number of studies explored the impact and significance of the pandemic in PD patients [5]. During the immunization period, different vaccines against SARS-CoV-2 (neutralizing, vector and *m*RNK) were available in the Republic of Serbia: inactivated virus vaccine (Sinopharm, Beijing, China), *m*RNA vaccine (Pfizer/BioNTech, New York, NY, USA) and adenovirus vector vaccines (Sputnik-V (Moscow, Russia) and Oxford-AstraZeneca (Cambridge, England, UK)) [6]. According to the World Health Organization, immunization was not mandatory but highly recommended for vulnerable groups, such as patients with different stages of chronic kidney disease (CKD), and particularly the dialysis population (HD and PD) and kidney transplant recipients (KTRs) [7,8,9]. The official recommendation was that the dialysis population should be immunized against SARS-CoV-2, except in situations where less than three months have passed since the previous COVID-19 infection [10]. Peritoneal dialysis patients, recognized as a population with a weakened immune response, are advised to receive a booster dose as well [11,12]. Despite that, both morbidity and mortality in this population of patients remained high throughout the pandemic. Standard risk factors, such as the presence of diabetes (DM), hypertension (HTN), obesity, and cardiovascular disease (CVD), could not account for the high prevalence of affected PD patients [12]. The presence of viable viral particles or viral RNA in the effluent of PD was considered for a while as a source of SARS-CoV-2 infection (laboratory workers, pathologists); however, this was shown in only a few studies whereas others were not able to confirm the result. The previous research that studied vaccination against COVID-19 in patients treated with PD was mainly based on *m*RNA vaccines, while there are little data on treatment with different types of vaccines [13].

Thus far, there are not many published papers on the impact of vaccination and previous COVID-19 infection, as well as the possible loss of anti-SARS-CoV-2 antibodies via the peritoneal membrane and effluent and their importance in the reinfection and survival of PD patients, which represent the main goals of this research.

## 2. Materials and Methods

### 2.1. Study Participants

The research was a longitudinal cross-sectional observational study that included 66 patients treated with PD at the University Clinical Center of Serbia, Clinic of Nephrology (UCCS-NFC). Patients were prospectively followed for 6 months, starting from December 2021. Blood samples were taken at 0 and 3 months, and patient outcomes (survival and occurrence of SARS-CoV-2 infection) were monitored for another 3 months. The entire study lasted for 6 months. At the same time, the control group (HC, n = 15) was formed based on hematological and biochemical parameters and anamnestic data. The research was approved by the Ethics Committee of UCCS, Decision Number 890/8 dated 21 December 2018. All subjects gave their written consent to participate in the research, while the research was conducted in accordance with the Declaration of Helsinki and the Ethical Guidelines for Medical and Health Research Involving Human Tests.

Only patients who did not have confirmed COVID-19 in the period of three months before the start of the study were included in the study. Patients who did not have peritonitis or clinical and laboratory signs of infection at the exit site of the peritoneal catheter a month before the start of the study were included. Patients who previously had a known hematological malignancy and were treated with immunosuppressive therapy were excluded from the study. The patients included in this study did not have autoimmune diseases, AIDS or tuberculosis. Blood was taken from all patients in the morning at 8.00 a.m. All examined patients had a negative virological status for hepatotropic viruses (anti-HCV and HbsAg) in the last 6 months and had aminotransferases (AST and ALT), gamma-GT and bilirubin (direct and indirect) within reference values. Acute infectious conditions were excluded by anamnesis and clinical examination, including examination of the exit site of the peritoneal catheter, as well as based on the absence of acute inflammation (parameters of the inflammatory syndrome within their reference values: leukocytes (Le), C-reactive protein (CRP), fibrinogen (Fib)). Patients with known hematological malignancies and those treated with immunosuppressive therapy were excluded from the study.

All PD patients were treated with standard glucose solutions, while 10 patients used glucose polymer (icodextrin) during the night shift. Sixty patients were treated with cyclic ambulatory peritoneal dialysis (CAPD) and six patients on automatic peritoneal dialysis (APD).

Medical histories were analyzed and an epidemiological questionnaire was conducted on previous COVID-19 infection, previous hospitalization in specialized COVID hospitals and vaccination status, as well as laboratory analyses from regular monthly controls. In addition, the results of all polymerase chain reaction (PCR) and rapid antigen test (RAT) assessments in the period of 12 months before the start of the study, conducted in the health institutions of the Republic of Serbia (from December 2020), were also analyzed, due to suspicions of illness as well as travelling abroad or necessary hospital treatment. All PCR tests (but not the RAT) performed in the Republic of Serbia are classified based on symptoms and signs of the disease as the main reason for testing.

### 2.2. Dialysis Parameters

The peritoneal equilibration test (PET) enables the assessment of transport characteristics of PM in patients treated for PD. It refers to the rate at which dissolved substances are transferred through the PM, until equilibrium is established for the given substance on both sides of the membrane in the circulation and infused dialysis solution. All patients had PET performed less than six months before the start of the study, following the recommendations of Cnosen et al. [14]. The adequacy of dialysis was assessed by total weekly urea clearance (Cu) and total weekly creatinine clearance (CCr) and was determined within 6 months before the beginning of the research, according to the procedure of [15]. The software package PD ADEQUEST 2.0. (Baxter Healthcare, Deerfield, IL, USA) was used to assess the quality of dialysis.

### 2.3. Laboratory Tests

Blood samples were taken from all participants (PD patients and control subjects) in the morning, after overnight fasting. For biochemical and serological analyses, blood was collected in tubes without anticoagulants, while whole blood samples were taken in Na-citrate tubes for fibrinogen and EDTA tubes for hematological tests. Biochemical and hematological parameters were determined at the Center for Medical Biochemistry of UCCS. Biochemical parameters (glucose, urea, creatinine, uric acid, total protein, albumin) and parathyroid hormone (PTH) were determined on an automated analyzer Architect ci8200 (Abbott Diagnostics, Wiesbaden, Germany) using routine test procedures. Whole blood counts were performed on an HmX hematology analyzer (Beckman Coulter, Inc., Brea, CA, USA).

The effluent sample was taken from the nightly dialysis exchange with 20 mL syringes. The sample was transferred to an empty test tube, centrifuged for 15 min at a speed of 3000 g and the supernatant (1.5 mL) transferred to a clean micro tube.

The presence of IgG antibodies against SARS-CoV-2 in sera and effluent was deter-mined using the commercial kit ELISA SARS-CoV-2 IgG (INEP, Belgrade, Serbia). The test is based on both spike and nucleocapsid viral antigens. The results are expressed as an index, calculated by the ratio of extinction of samples over the extinction of the calibrator. The cut-off value for the test is defined on the basis of the ROC curve. The results are classified as negative if the index is less than 15, and the borderline zone is from 15 to 20, while values above 21 are considered as positive.

### 2.4. Statistical Analysis

The normality of data distribution was tested using the Kolmogorov–Smirnov test. For data following a bell-shaped distribution, the results are presented as the mean ± standard deviation, whereas the data diverging from it are presented as the median ± interquartile range. The χ^2^ test or Fisher’s exact test was used to compare categorical variables. The comparison between the studied groups was conducted using Student’s *t*-test or the Mann–Whitney U test, depending on the normality of the data distribution. The Pearson test was used to examine the correlations of the obtained parameters. Statistical significance was defined as a *p* value < 0.05. Statistical analysis was performed in the SPSS v.18 program (Chicago, IL, USA).

## 3. Results

### 3.1. Basic Clinical and Laboratory Parameters

The average age of the studied PD patients population was 61.1 ± 14.2 (minimum: 27, maximum: 89 years). The average duration of dialysis treatment was 38.8 ± 38.4 (minimum: 4, maximum: 168 months). The average age of the examined group of HC was 57.1 ± 4.2 (minimum: 40, maximum: 62 years). Patients treated with PD were significantly older compared to their peers in the HC group (61.1 ± 14.2 vs. 57.1 ± 4.2, *p* < 0.001). Hematological and biochemical parameters in the population of patients treated with peritoneal dialysis, as well as their basic dialysis characteristics, are presented in Table 1.

The total protein content decreased during the follow-up period, but significance was not reached (*p* = 0.061). The values of Alb, Fib, Hb, CRP and Plt did not change over the follow-up period (Table 2).

### 3.2. COVID-19 Vaccination and Natural Immunization Status

A total of 38 (57.6%) PD patients underwent primary vaccination (PV) with at least one dose of anti-SARS-CoV-2 vaccine, while the remaining 28 (42.4%) did not. As PV, in the group of vaccinated PD patients, 32 (84.2%) received Sinopharm, 5 patients (13.0%) Pfizer–BioNTech and 1 patient (2.6%) Sputnik-V. The third, booster dose (secondary vaccination, SV), was received in total by only 22 PD patients: heterologous booster (a combination of Sinopharm and Pfizer-BioNTech) was received by 9 patients (23.7%), and 13 patients (34.2%) received three doses of the vaccine homologous booster dose, while 15 patients (39.5%) remained on PV. The majority of the group of vaccinated PD patients received Sinopharm, followed by the combined vaccine of Sinopharm + Pfizer-BioNTech or Pfizer-BioNTech and Sputnik-V (Figure 1).

In our group of examined PD patients, 9 patients (9.1%) were infected with COVID-19 before the start of the study, but in a time interval longer than 3 months before the start of the study. In the period of 3 months between the two measurements of the anti-SARS-CoV-2 antibody titer, PD patients did not have peritonitis. In the same period, they did not have a SARS-CoV-2 infection, and they were not vaccinated against COVID-19 during the entire follow-up period of 6 months.

In the HC group, 15 people were vaccinated (period longer than 45 days), and all of them were vaccinated with the *m*RNA vaccine.

### 3.3. Anti-SARS-CoV-2 Antibody Levels and Their Kinetics in PD Patients

There was a significant difference in the observed IgG level between patients treated with PD and the control group (*p* < 0.001) (Table 3). In patients treated with PD during the follow-up period, the level of IgG antibodies in the serum did not change significantly (*p* = 0.249). In the control group, during the monitoring period, the level of IgG antibodies significantly decreased (*p* = 0.001).

The level of IgG antibodies in the serum was associated with vaccination status and acquired immunity (AI); namely, it was significantly different at the first measurement (*p* = 0.003) and at the second measurement (*p* = 0.002) (Table 4). Subjects with acquired immunity had a significantly higher level of IgG antibodies on the first measurement compared to vaccinated patients (*p* = 0.006), as well as compared to the group of patients who did not have clinical presentation of the disease (0.001). The trend also continued at the second measurement (*p* = 0.008, i.e., *p* = 0.001).

In our research, anti-SARS-CoV-2 antibodies were also verified in the effluent. Their levels did not change significantly during the monitoring period (*p* = 0.979) (Table 5).

Levels of serum IgG antibodies did not change significantly during the follow-up period in either vaccinated or non-vaccinated patients (*p* = 0.592, respectively *p* = 0.248), nor did it differ in relation to the vaccination status at the first measurement (*p* = 0.900) or at the second measurement (*p* = 0.560) (Table 6). IgG antibodies in the effluent did not change significantly during the follow-up period in either vaccinated or non-vaccinated patients (*p* = 0.373, respectively *p* = 0.498). The level of IgG antibodies in the effluent did not differ significantly in relation to the vaccination status at the first measurement (*p* = 0.509) or at the second measurement (*p* = 0.957) (Table 6).

PD patients are further divided into two groups according to age: Group 1 (<60 years) and Group 2 (>60 years). The level of IgG antibodies in the serum did not change significantly during the follow-up period in patients in Group 1 (*p* = 0.875), nor in patients in Group 2 (*p* = 0.192). In addition, the serum IgG levels did not differ in relation to the age of the patient at the first measurement (*p* = 0.860) or at the second measurement (*p* = 0.235) (Table 7). The same was noted for the effluent levels of IgG. IgG antibodies in the effluent did not change significantly during the follow-up period in those younger than 60 years old in Group 1 (*p* = 0.223), and in those older than 60 years old (*p* = 0.476). The level of IgG antibodies in the effluent did not differ significantly in relation to the age of the patients at the first measurement (*p* = 0.519) or at the second measurement (*p* = 0.691) (Table 7).

There was no significant correlation between the level of IgG antibodies in the serum and the age of PD patients (*r* = −0.102, *p* = 0.413). Nevertheless, there is a significant correlation between the level of IgG antibodies in the serum both at the first (*r* = 0.531, *p* = 0.042) and at the second measurement (*r* = 0.522, *p* = 0.046) and the age of the subjects of the control group (Figure 2).

In this study, four PD patients had a positive SARS-CoV-2 PCR test during the follow-up period. All infected patients were not immunized and all had a mild clinical picture. In our study, during six months of follow-up, five patients were switched to HD (7%). During the same period, there were nine lethal outcome (13%) from a cardiovascular event. Of the deceased, two patients were vaccinated, but had no detectable antibodies in either of the two measurements. Of the total number of patients who died, seven had no detectable anti-SARS-CoV-2 IgG antibodies, while two patients had anti-SARS-CoV-2 IgG antibodies at the second measurement, without previous immunization and confirmed infection.

## 4. Discussion

The presented research was conducted to examine the concentration, as well as the dynamics of anti-SARS-CoV-2 antibodies in serum and effluent in PD patients who had previously suffered from a COVID-19 infection and in patients who were vaccinated before a follow-up period longer than 1 month. It also aimed to investigate the association of anti-SARS-CoV-2 IgG concentration in serum and effluent during the six-month follow-up period. The results were compared with analyses of a HC group that was matched for sex, but younger in age. This research confirmed the existence of anti-SARS-CoV-2 IgG in the effluent. The results of others highlighted mostly adverse reactions to vaccination in PD patients [7,16]. They are more often registered in women, in the form of a local reaction and pain at the site of application. None of the mentioned patients had serious adverse reactions that required hospital treatment, which is in accordance with the results reported herein. Unlike the previous studies [11,16], our research extended the follow-up period of vaccination outcomes, and included the determination of anti-SARS-CoV-2 IgG levels for 3 months, while the patients were monitored for 6 months. During the mentioned period, the dominant strain in the Republic of Serbia, but also throughout the world, was the omicron variant of SARS-CoV-2.

Patients with ESKD show an increased tendency to develop severe forms of infection with COVID-19 with a lethal outcome compared to the healthy population, which is why it is still necessary to shed light on the mechanisms that predominate towards an unfavorable outcome in this group of patients [17]. The significance of this research is also greater due to the fact that PD as a modality of KRT is a less frequently applied form of treatment compared to the much more prevalent HD both globally and in the Republic of Serbia, and the favorable circumstances of conducting the dialysis procedure at home became more obvious during the pandemic. This modality potentially enables less frequent exposure of patients to viral particles, among other things, due to less frequent visits to health care facilities. It is known that this modality of KRT also enables better cardiovascular stability, since it has been shown that patients with DM, HTN, obesity and heart failure with reduced ejection fraction (HFrEF and HFmrEF) represent a group that is at increased risk of death from COVID-19 infection [18]. Since the representation of patients with DM is up to 70% in the PD population, and cardiovascular comorbidity is present up to 80% [11], as well as the fact that chronic inflammation and sarcopenia are more common in this group of patients, PD patients are predisposed to more severe forms of infection. Considering the development of new technologies (remote monitoring and control of patients), the tendency to improve the biocompatibility of dialysis solutions and a lower epidemiological risk, an increase in the percentage of patients treated with this KRT modality is expected. This is of particular interest when considering pandemics of infectious disease, less frequent visits to health facilities, and collective transport of patients to one of the HD centers [19,20].

Around half (56.1%) of patients in the PD group were vaccinated, which is significantly lower than in the study conducted by [19], where 95% of PD patients were immunized. The reason for postponing immunization in 5% of patients in the aforementioned study was mainly safety concerns due to the new *m*RNA vaccine. Previous research conducted among the HD population reported similar reasons for indecision and delay in vaccination, while kidney transplant recipients (KTRs), regardless of the type of vaccine against COVID-19 infection, and if they were vaccinated in the highest percentage of *m*RNA, did not produce a high titer of antibodies regardless of the number of vaccines [9,21,22,23,24]. In our investigated population, the lower representation of immunized patients can also be explained by the fact that nine patients who were infected with SARS-CoV-2 believed that they had achieved adequate protection against potential reinfection. Due to the nature of performing dialysis treatment at home, one part of the patients believed that isolation was a sufficient measure of prevention against infection, which is why they postponed immunization. Due to the lack of data on the adverse effects of the SARS-CoV-2 vaccine in the PD patient population, a safety profile study would be of great importance. A study on the population of HD patients showed that the *m*RNA vaccine leads to the creation of a higher titer of Ab, but also to a faster weakening of the immune response, and research by Brković et al. showed a significant impact of vaccination on the survival of the HD population compared to unvaccinated patients [25,26].

In the PD group, the level of anti-SARS-CoV-2 IgG antibodies in serum did not change significantly during the follow-up period, while in the HC group the decrease was significant (*p* = 0.001). Subjects with immunity acquired after previous infection had a significantly higher level of anti-SARS-CoV-2 IgG antibodies at the first measurement compared to vaccinated patients (*p* = 0.006), as well as to the group of patients who did not have a clinical presentation of the disease (*p* = 0.001). Zheng et al. [16] compared the level of IgG antibodies after immunization between PD and HD patients and showed that PD patients produced a higher titer of anti-SARS-CoV-2 IgG. However, they did not examine the dynamics at two time points, nor did they determine the eventual loss of antibodies through the effluent. In our study group, the presence of anti-SARS-CoV-2 IgG antibodies was verified in the effluent. Their levels decreased insignificantly during the monitoring period (*p* = 0.0979).

A shorter dialysis period is less often associated with complications, as patients have longer preserved residual renal function, which is why the need for follow-up examinations and hospital treatment is less frequent. That is significant for the healthcare system, which is completely focused on COVID-19 patients. Elderly patients are at greater risk of an adverse outcome from COVID-19 [8]. Given that life expectancy has increased and that the frequency of ESKD has increasingly shifted towards the geriatric population, this is another piece of information that is important for starting dialysis treatment with PD.

It has long been known that ESKD patients have an altered concentration of most biochemical and hematological parameters, except for lipid profile, leukocyte and blood platelet count, which we also observed in the present research, while the satisfactory quality of dialysis indicated the competence of the PD as a KRT method [27]. In our group of examined PD patients, nine patients (9.1%) were infected with COVID-19 before the start of the study, but in a time interval longer than 3 months before the start of the study. In this study, slightly higher values of biomarkers of acute inflammation (CRP and Fib), were observed in four patients who had a positive COVID PCR test during the follow-up period. All infected patients were non-immunized, and they all had a mild clinical presentation. The majority of patients had their primary infection during the period of dominance of the δ strain, while the majority of immunized patients had their primary infection during the omicron strain. Htay et al. [28] state that the majority of PD patients in their study who underwent infection with the δ strain were immune to omicron, while only immunized individuals were more likely to suffer from omicron, which does not correspond to our results. Given the small number of patients, the result should be taken with caution. In patients treated with PD, the parameters of chronic inflammation (accelerated erythrocyte sedimentation rate, ferritin and d-dimer) may remain elevated for a longer period of time after prolonged SARS-CoV-2 infection (longer than 3 months), which would correspond to chronic inflammation [29].

It is known that as a result of dialysis treatment, there is a loss of protein through the peritoneal membrane as well as possible malnutrition, which conditions the occurrence of sarcopenia [30]. In the group of patients included in this study, the decrease in protein content was insignificant. The values of specific proteins (Alb, Fib, Hb, CRP) but also blood Plt did not change significantly during the follow-up period. In the group of PD patients, a total of 38 patients were vaccinated. The largest number of patients received the Sinopharm vaccine (Verocell), which at the beginning of the immunization period was the most available in the Republic of Serbia, and which was used for immunization of the majority of the population. A heterologous booster (a combination of Verocell and *m*RNA vaccine) was received by nine patients, and it was the most common type of immunization in the country. According to literature data, this vaccine combination creates the strongest immune response [31]. Ten patients received three doses of homologous dead vaccine boosters. In most countries of the Western world, a homologous booster with three doses of *m*RNA vaccine has been applied [31].

Given that previous publications indicate that the immune response to vaccination status weakens after the sixth decade of life [32], in our study PD patients were classified into two groups according to age. In contrast to the reports of [33], we found no significant association between the level of anti-SARS-CoV-2 IgG antibodies in serum and the age of the patient. In our research, correlation analysis confirmed the significance of the loss of anti-SARS-CoV-2 IgG antibodies through the effluent in the control measurement. Given that there are no published papers on the loss of anti-SARS-CoV-2 IgG antibodies across the peritoneal membrane, these results can be considered the first of their kind and are characteristic of an older population. At the beginning of the experimental setup, as well as throughout the follow-up period, the dominant strain of SARS-CoV-2 was omicron. The same strain has also been the dominant strain worldwide for the last 18 months. In earlier studies that examined the impact of vaccination, one of the exclusion criteria for patient participation in the study was a previous history of SARS-CoV-2 infection (nucleocapsid positive serological) or that they had been vaccinated against other viral diseases in a period shorter than one year, as well as belonging to the group of patients on immunosuppressive therapy [33].

Herein, we included all PD patients, regardless of their history of SARS-CoV-2 infection and vaccination status, but none of the patients had previously taken immunosuppressive therapy. In the period of occurrence of omicron subvariants, we investigated the protective role of anti-SARS-CoV-2 antibodies in PD patients acquired by previous vaccination and hybrid immunity ((HI) previous infection and vaccination). The efficacy of protection against symptomatic omicron infection was assessed by testing each symptomatic individual with the RAT and it was shown that vaccination was successful against symptomatic infection, which is inconsistent with earlier results [28]. They indicated that vaccination was successful against a severe form of the disease in the first months of the new strain (emergency hospitalization due to the need for oxygen therapy, acute respiratory distress syndrome (ARDS), treatment in intensive care units or fatal outcome), which speaks in favor of the effectiveness of vaccination, natural immunization or HI against the omicron variant. Htay et al. [28] stated that the majority of PD patients in their study who had undergone δ strain infection were immune to omicron, while only immunized persons were more likely to suffer from omicron, which does not correlate with our results.

In our study, during six months of follow-up, five patients were transferred to HD (7%). All patients were previously immunized, and no anti-SARS-CoV-2 IgG antibodies were detected in two patients. In the same period, there were nine deaths (13%), two of which were vaccinated, but had no detectable antibodies in either of the two measurements. Of the total number of patients that have died, seven had no detectable anti-SARS-CoV-2 IgG antibodies, while two patients had anti-SARS-CoV-2 IgG antibodies at the second measurement, without previous immunization. The mentioned patients were not tested with the RAT due to the absence of a febrile condition, which was the criterion for testing at the given time according to the recommendation of the National Coordinating Body for Pandemic Management in the Republic of Serbia, which is why the result should be taken with caution. Our results are inconsistent with the above data. Previous infection with a non-omicron variant (α, β, γ and δ) was associated with a 100% reduction in new infection in the first three months of the Stealth Omicron subvariant [32], which correlates with our study considering that all six patients who had the δ strain and were not immunized also had a reinfection with the omicron strain.

Primary vaccination (PV) series without a previous COVID-19 infection had a protective role against the omicron variant. Most of the patients received two doses of the vaccine in a time interval longer than 45 days, before the appearance of the new strain. The protective effect is also reflected in the fact of a short period of immunization, given that literature data speak of a short-term protective effect of PV against omicron strains and a more permanent protection against natural infection (if more than 6 months have passed). The use of homologous and heterologous booster doses was associated with a 100% reduced risk of infection/reinfection in our study, which deviates from the literature data that speak of a protective effect in 60% of patients with HI, and 50% in patients who acquired only natural immunity [22,33]. During the period when experiment was conducted, the dominant strain in the Republic of Serbia was Stealth, and no significant differences were observed between the effects of the dead, inactivated or *m*RNA vaccine. The results should be taken with a grain of salt considering the small number of subjects in our study who received a booster dose of the vaccine. A high efficiency can be attributed to a point when booster dose was received, which was in a period from 45 to 75 days. In our study, three patients had HI (previous infection and vaccination), which was shown to be completely effective against omicron strain reinfection. This finding indicates that the HI in the group of PD patients has a significant efficiency. Given that previous infection reduces the risk of reinfection by 50%, and subsequent vaccination by 60% [33], our results should be taken with caution considering the small number of patients with HI (4%). The effect of HI needs to be further investigated. Each form of protection showed large differences in the occurrence of symptoms and the form of the disease, but each of them also showed significant protection against hospitalization, the need for oxygen therapy, the development of ARDS or death related to COVID-19 infection; in our research, an effectiveness of almost 100% was found. This result indicates that any form of previous immunity (acquired by previous infection, vaccination or hybrid) is associated with strong and significant immune protection against hospitalization and death related to COVID-19 in the studied group, which is in accordance with literature data [34,35].

## 5. Conclusions

In our research, 57.6% of patients treated with peritoneal dialysis (PD) were vaccinated against SARS-CoV-2 infection. The largest number of patients (81.6%) was immunized with the Sinopharm vaccine, which was also the most prevalent in the Republic of Serbia at the start of immunization. None of the patients had serious side effects from the vaccination. During the monitoring period, the level of anti-SARS-CoV-2 IgG antibodies in the group of PD patients had an increasing trend, and in the group of healthy controls (HC), the level of anti-SARS-CoV-2 IgG antibodies had a decreasing trend. In the group of vaccinated PD patients, anti-SARS-CoV-2 IgG antibodies had an increasing trend in both serum and effluent, in contrast to non-vaccinated patients, where they decreased in the effluent regardless of the trend of increase in the serum, but significance was not reached. In the HC group, the level of anti-SARS-CoV-2 IgG antibodies in the serum was higher at both measurements, while the level was not significantly different in the two age-groups (older and younger than 60 years). There was no significant correlation of anti-SARS-CoV-2 IgG antibody levels in the serum and effluent with hematological parameters and biochemical analyses; however, IgG antibodies in the serum and effluent correlated with each other. The immunized patients did not acquire the omicron strain infection, while the patients who underwent infection with the δ strain without immunization all had a reinfection with the omicron strain, but without severe clinical presentation and they did not require hospital treatment.

## Figures and Tables

**Figure 1 vaccines-12-00135-f001:**
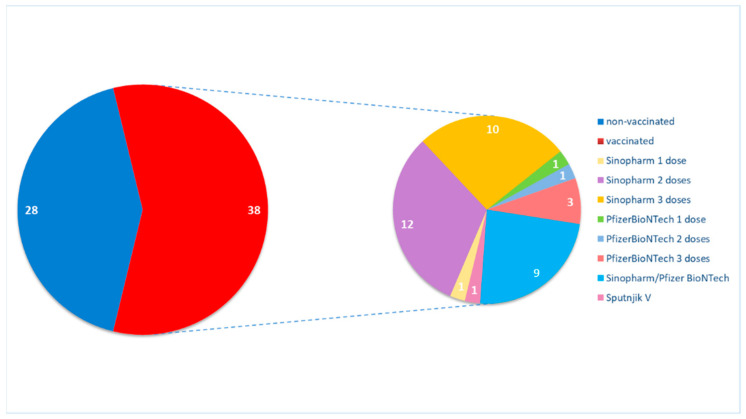
COVID-19 vaccination status in PD patients. The number in the graph represents the number of study subjects.

**Figure 2 vaccines-12-00135-f002:**
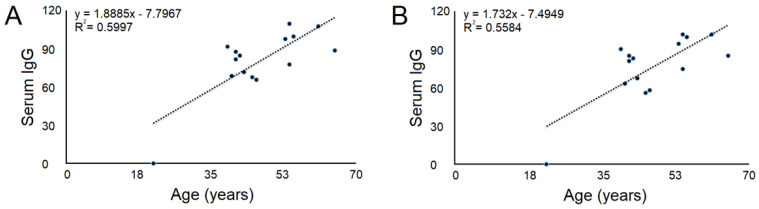
Correlation of IgG antibodies at the first measurement (**A**) and at the second measurement (**B**) with age in subjects of the control group.

**Table 1 vaccines-12-00135-t001:** Basic hematological, biochemical and dialysis parameters in the studied PD group at the beginning of the study.

Parameters, N = 66	Mean ± SD	Median
iPTH, ng/mL	412.86 ± 246.5	400.5
Hb, mean ± SD	105.42 ± 12.8	103
Plt, mean ± SD	268.33 ± 88.44	251
Glu, mean ± SD	6.23 ± 2.77	5.4
Ur, mean ± SD	15.1 ± 4.52	15.35
Cr, mean ± SD	726.09 ± 209.17	715.5
Alb g/L, mean	35.08 ± 4.38	36
TP g/L, mean	63.83 ± 6.63	64
Fib g/L, mean	4.97 ± 0.89	4.95
CRP g/L	7.24 ± 9.79	3.9
Chl (mmol/L)	5.04 ± 1.22	4.78
Tg (mmol/L)	1.76 ± 0.91	1.56
UF (L/day)	1.19 ± 0.76	1.1
RU (L/day)	0.92 ± 0.78	0.9
Kt/V	2.44 ± 0.61	2.5
CCr (L/week)	87.89 ± 33.88	82
*PET-gly*	0.46 ± 0.16	0.43
PET-Cr	0.67 ± 0.12	0.67

UF—ultrafiltration; RU—residual urine; Kt/V—urea clearance; CCr—weekly creatinine clearance; PET—peritoneal equilibrium test for glucose and creatinine; TP—total protein; Alb—albumin, Fib—fibrinogen; Hb—hemoglobin; Plt—platelets; CRP—C-reactive protein; glu—glucose; Ur—serum urea, Cr—serum creatinine; Chl—total cholesterol; Tg—triglycerides; iPTH—parathyroid hormone.

**Table 2 vaccines-12-00135-t002:** Hematological and biochemical parameters in two measurements in PD patients: Measurement 1—values at the beginning of the study, Measurement 2—values obtained 3 months after *t*-test ^1^ for dependent samples.

Parameters	Measurement 1	Measurement 2	*p*-Value ^1^
TP	63.83 ± 6.49	61.51 ± 10.84	0.061
Alb	35.08 ± 4.38	34.52 ± 6.21	0.66
Fib	4.97 ± 0.89	5.20 ± 0.88	0.156
Hb	105.42 ± 12.8	106.65 ± 10.88	0.262
Plt	268.33 ± 88.44	281.35 ± 91.22	0.184
CRP	7.24 ± 9.79	17.54 ± 53.21	0.144

TP—total protein; Alb—albumin; Fib—fibrinogen; Hb—hemoglobin; Plt—platelets; CRP—C-reactive protein; *p* first vs. second measurement.

**Table 3 vaccines-12-00135-t003:** IgG antibody values in the serum of patients treated with PD (unrelated to their vaccination status) and the healthy control group (HC). *p*-value ^1^—difference between PD and HC; *p*-value ^2^—difference between measurement 1 and 2 within each group; *—significance < 0.05.

Parameters	PD	HC	*p*-Value ^1^
Measurement 1	27.80 ± 35.17	80.33 ± 26.26	**<0.001 ***
Measurement 2	33.50 ± 36.07	73.33 ± 24.26	**<0.001 ***
*p*-Value ^2^	0.249	**0.001 ***	

PD—patients treated with peritoneal dialysis; HC—healthy control group. Significance is in bold and marked with an asterisk (*).

**Table 4 vaccines-12-00135-t004:** Serum IgG antibody values in patients treated with PD depending on vaccination status and disease. V—vaccinated; AI—acquired immunity; HI—hybrid immunity. *p*-value ^1^—difference between V, AI and HI; *p*-value ^2^—difference between measurement 1 and 2 within each group; *—significance < 0.05.

Parameters	V (n = 35)	AI (n = 6)	HI (n = 3)	*p*-Value ^1^
Measurement 1	39.03 ± 38.32	70.31 ± 30.99	20.14 ± 35.75	**0.003 ***
Measurement 2	41.53 ± 35.95	71.69 ± 27.67	24.95 ± 35.70	**0.002 ***
*p*-Value ^2^	**0.001 ***	**0.006 ***	**0.008 ***	

**Table 5 vaccines-12-00135-t005:** IgG antibody values in serum and effluent in patients treated with PD. *t*-test ^1^ for dependent samples.

Parameters, N = 66	Measurement 1	Measurement 2	*p*-Value ^1^
IgG serum	27.80 ± 35.17	33.50 ± 36.07	0.249
IgG effluent	5.20 ± 12.75	5.09 ± 9.62	0.979

**Table 6 vaccines-12-00135-t006:** Antibody values in relation to vaccination status in patients treated with PD. *p*-value ^1^—difference between vaccinated and non-vaccinated; *p*-value ^2^—difference between measurement 1 and 2 within each group.

Antibodies	Vaccinated, N = 38	Non-Vaccinated, N = 28	*p*-Value ^1^
IgG serum t0	31.03 ± 34.92	23.80 ± 35.78	0.90
IgG serum t1	35.19 ± 34.62	31.40 ± 38.41	0.560
*p*-Value ^2^	0.592	0.248	
IgG effluent t0	4.04 ± 12.09	6.13 ± 13.38	0.509
IgG effluent t1	5.52 ± 11.03	4.74 ± 8.48	0.957
*p*-Value ^2^	0.373	0.498	

**Table 7 vaccines-12-00135-t007:** Values of the level of IgG antibodies in serum and effluent in relation to the age of the patient. Group 1 under 60 years old, Group 2 over 60 years old. *p*-value ^1^—difference between Group 1 and Group 2; *p*-value ^2^—to vs. t1.

Antibodies	Group 1, N = 24	Group 2, N = 42	*p*-Value ^1^
IgG serum t0	27.00 ± 38.98	28.32 ± 33.07	0.860
IgG serum t1	30.41 ± 36.41	35.50 ± 36.26	0.5235
*p*-Value	0.875	0.192	
IgG effluent t0	2.27 ± 7.41	7.09 ± 5.05	0.519
IgG effluent t1	4.14 ± 8.04	5.71 ± 10.58	0.691
*p*-Value ^2^	0.223	0.476	

^1^ *t*-test for independent samples, ^2^ *t*-test for dependent samples.

## Data Availability

Data available on request due to restrictions.

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
