# Peer review of "The Importance of Natural and Acquired Immunity to SARS-CoV-2 Infection in Patients on Peritoneal Dialysis"

_vaccines, 2024, doi:10.3390/vaccines12020135_

Round 1

Reviewer 1 Report

Comments and Suggestions for Authors

1) Readability should be improved: a) Grammar, e.g. Abstract: This study demonstrated ... -> demonstrates, Table 1 'Parametars' ->'Parameter '. Table 2 '2st' values -> 2nd values. b) Clarity Introduction: PD ... is one of the three complementary methods ... -> the authors mention HD as a second method, but the third remains elusive. c) Abbreviations in Table 2 are not complete  ->CRP missing. d) Partially redundant tables e.g. Tables 1 and 2, should be merged into one e) Information given in the tables should be reconsidered e.g. why does the reader need to know min and max values of Table 1?

2) The Discussion should limit its content to the data given in the Results section e.g. I could not find any connection between the information given in the second paragraph and any of the Results topics. 

3) Methods topic 2.2 Vaccination status  is not a method, but a basic parameter of the patient population and belongs in the Results section.  

4) The paper summarises laboratory data of immunoglobulin levels in patients, but the implications or consequences are not clear to me. The authors should try to describe, what we now know more about and what we would not know without these data.

Comments on the Quality of English Language

I understood the English well, however, there are grammatic and spelling issues. 

Type at the beginning of Materials and Methods 'The The'

Author Response

1) Readability should be improved: a) Grammar, e.g. Abstract: This study demonstrated ... -> demonstrates, Table 1 'Parametars' ->'Parameter '. Table 2 '2st' values -> 2nd values. b) Clarity Introduction: PD ... is one of the three complementary methods ... -> the authors mention HD as a second method, but the third remains elusive. c) Abbreviations in Table 2 are not complete ->CRP missing. d) Partially redundant tables e.g. Tables 1 and 2, should be merged into one e) Information given in the tables should be reconsidered e.g. why does the reader need to know min and max values of Table 1?

-Thank you for your suggestions. Authors have adopted the recommendations to improve the accompanying technical content that follow the subject matter discussed.  

2) The Discussion should limit its content to the data given in the Results section e.g. I could not find any connection between the information given in the second paragraph and any of the Results topics. 

  • - Authors find this information useful in the case of future pandemics, as well as future workload organization of PD and HD centers to tackle the changed workflow for the benefit of the patients. Authors, however, can make additional edit if needed for the sake of clarity.
  •  

3) Methods topic 2.2 Vaccination status is not a method, but a basic parameter of the patient population and belongs in the Results section.

- Thank you for your comment. Authors have adopted your valuable suggestion and corrected the oversight.  

4) The paper summarises laboratory data of immunoglobulin levels in patients, but the implications or consequences are not clear to me. The authors should try to describe, what we now know more about and what we would not know without these data.

Thank you for your comment. Authors find that, based on our study, immunization among PD patients have significantly reduced the possibility for a severe clinical presentation and hospitalization as a result. This premise is particularly significant, as it tackles vulnerable population. In addition, we have observed that older PD patient population had produced higher levels of anti SARS CoV-2 IgG antibodies, compared to younger PD patient population, which could be of importance in terms of administration of vaccine boosters over periods longer than 6 months for which they are necessary further research!

Reviewer 2 Report

Comments and Suggestions for Authors

1.     Did all the patients included in present study has any diseases related to immunological disorders? Such as auto-immune diseases, AIDS, TB, etc.

2.     Patients treated with PD were significantly older compared to their peers in the HC group in present study. As we know, the older health population usually showed weaker B and T response to antigen stimulation and lower IgG level compared to the adult or adolescence. And in present study, we noticed the PD patients has lower IgG titers compared to the HC group, so how to exclude the interference of age on immune response? Or the authors should compare the IgG titer of PD to published older health populations. Although this confusion has been somewhat salvaged by Table 7, it failed to provide direct proof for comparing the IgG level between patients and health population.

3.     The authors were suggested to describe the time intervals between the last injection and 1st-values at the beginning of the study,as the titers of IgG were increased from 1st-values to 2ed-values.

4.     I am difficult to follow the results of Table 5 and Table 6, as the author documented, there are 38 vaccinated patients of PD, so why the remained non-vaccinated patients could elicit IgG against SARS-CoV-2, did they were infected by virus previously?  

5.     In figure1, did the r2=0.5597 or 0.5584 means significant correlation? As this correlation was difficult to be understand: the older population will produce more antibody during vaccination?

6.     In abstract, the authors declared that “Immunized patients did not acquire the omicron strain infection, while patients who 33 underwent the δ strain without immunization all had reinfection with the omicron strain, but 34 without severe clinical presentation and did not require hospital treatment.” No experimental results support this conclusion.

Author Response

Did all the patients included in the present study have any diseases related to immunological disorders? Such as auto-immune diseases, AIDS, TB, etc.

- None of the patients observed in this study have had any immunological disease, such as AIDS, TB, etc. We have elaborated on this fact in the study inclusion criteria.

2. Patients treated with PD were significantly older compared to their peers in the HC group in the present study. As we know, the older healthy population usually showed a weaker B and T response to antigen stimulation and a lower IgG level compared to the adult or adolescent. And in the present study, we noticed that the PD patients had lower IgG titers compared to the HC group, so how to exclude the interference of age on immune response? Or the authors should compare the IgG titer of PD to published older healthy populations. Although this confusion has been somewhat salvaged by Table 7, it failed to provide direct evidence for comparing the IgG level between patients and the healthy population.

  • Thank you for your comment. Authors find that, based on our study, immunization among PD patients significantly reduced the possibility for a severe clinical presentation and hospitalization. This is particularly significant, as it tackles vulnerable population. In addition, older PD patient population produced higher levels of anti-SARS CoV-2 IgG antibodies, compared to the younger PD patient population, which could be of importance in terms of administration of vaccine boosters over periods longer than 6 months for which they are necessary further research!

3. The authors were suggested to describe the time intervals between the last injection and 1st-values at the beginning of the study, as the titers of IgG were increased from 1st-values to 2nd-values.

- The authors have entered and dully described the information needed. 

4. I am difficult to follow the results of Table 5 and Table 6, as the author documented, there are 38 vaccinated patients of PD, so why the remaining non-vaccinated patients could elicit IgG against SARS-CoV-2, did they were infected by virus previously?

- Authors have assumed that those patients had asymptomatic infection; therefore they were not tested with RAT or PCR.

5. In figure 1, did the r2=0.5597 or 0.5584 mean significant correlation? As this correlation was difficult to understand: the older population will produce more antibody during vaccination?

- That is correct. Older PD patients have actually produced more anti SARS CoV-2 IgG antibodies during the immunization period. 

6. In abstract, the authors declared that "Immunized patients did not acquire the omicron strain infection, while patients who 33 underwent the δ strain without immunization all had reinfection with the omicron strain, but 34 without severe clinical presentation and did not require hospital treatment.” No experimental results support this conclusion.

-Authors provided data in the Results section stipulating the number of patients that previously had δ strain, and how many patients had omicron strain. 

Reviewer 3 Report

Comments and Suggestions for Authors

In this work, Baralic et al. studied the impact of infection and vaccination in 66 patients treated with PD and their outcomes during a 6-month follow-up. It was observed that the level of anti-SARS-CoV-2 IgG antibodies in PD patients had an increasing trend in serum. Moreover, in the group of vaccinated patients with PD, anti-SARS CoV-2 IgG antibodies showed an increasing trend in both serum and effluent, contrary to non-vaccinated patients, where antibodies decreased in effluent regardless of the trend of increase in serum. In my opinion, the study is interesting and can be accepted for publication after minor revision. The authors must address the following concerns properly:

1) The introduction and experimental (testing of samples) sections should be improved further as the authors have missed some important articles on SARS-COV-2 testing/detection and its wider impact such as https://iopscience.iop.org/article/10.1088/2043-6262/aceda9/meta; https://www.sciencedirect.com/science/article/pii/S2468519420300665

2) What is the rationale behind taking the sample size of 66  PD patients?

3) Page 2, line 71; please correct the repetition.

4)  The study related to specific vaccines and an equal number of patients should give a better idea.

5) The quality of Fig.1 is very poor. Please enhance its resolution.

6) Can the obtained results for 6 months be extrapolated beyond this?

Author Response

n this work, Baralic et al. studied the impact of infection and vaccination in 66 patients treated with PD and their outcomes during a 6-month follow-up. It was observed that the level of anti-SARS-CoV-2 IgG antibodies in PD patients had an increasing trend in serum. Moreover, in the group of vaccinated patients with PD, anti-SARS CoV-2 IgG antibodies showed an increasing trend in both serum and effluent, contrary to non-vaccinated patients, where antibodies decreased in effluent regardless of the trend of increase in serum. In my opinion, the study is interesting and can be accepted for publication after minor revision. The authors must address the following concerns properly:

1) The introduction and experimental (testing of samples) sections should be improved further as the authors have missed some important articles on SARS-COV-2 testing/detection and its wider impact such as https://iopscience.iop.org/article/10.1088/2043-6262/aceda9/meta;

https://www.sciencedirect.com/science/article/pii/S2468519420300665

- -Thank you for your comments and suggestions. Authors will expand the indicated sections  in order to thoroughly support the narrative presented in the work.  

2) What is the rationale behind taking the sample size of 66 PD patients?

-All PD patients from the dialysis center NFC UCCS (total of 66) were included in the study, except the ones excluded based on the exclusion criteria. 

3) Page 2, line 71; please correct the repetition.

- Authors corrected the technical error. Thank you for your input.  

4) The study related to specific vaccines and an equal number of patients should give a better idea.

Thank you for your comment. The study included vaccines that were available and used in our country during the time of the study. Authors have listed this information in Figure 1. 

5) The quality of Fig. 1 is very poor. Please enhance its resolution.

- Authors have corrected the Figure 1 and provided the image in higher resolution in order to provide better visibility of information. 

6) Can the obtained results for 6 months be extrapolated beyond this? 

- Thank you for the posted comment. We continued to monitor the PD patients and the effect of vaccination for a period of 12 months, the results of which we expect, but the morbiditz of this group pf patients was significantly less during the extended studies that were immunized. Authors find that, based on our study, immunization among PD patients significantly reduced the possibility for a severe clinical presentation and hospitalization. This is particularly significant, as it tackles vulnerable population. In addition, older PD patient population produced higher levels of anti-SARS CoV-2 IgG antibodies, compared to the younger PD patient population, which could be of importance in terms of administration of vaccine boosters over periods longer than 6 months for which they are necessary further research!

Round 2

Reviewer 2 Report

Comments and Suggestions for Authors

All my questions have been resolved, I have no further comment.